# OATD-02 Validates the Benefits of Pharmacological Inhibition of Arginase 1 and 2 in Cancer

**DOI:** 10.3390/cancers14163967

**Published:** 2022-08-17

**Authors:** Marcin Mikołaj Grzybowski, Paulina Seweryna Stańczak, Paulina Pomper, Roman Błaszczyk, Bartłomiej Borek, Anna Gzik, Julita Nowicka, Karol Jędrzejczak, Joanna Brzezińska, Tomasz Rejczak, Nazan Cemre Güner-Chalimoniuk, Agnieszka Kikulska, Michał Mlącki, Jolanta Pęczkowicz-Szyszka, Jacek Olczak, Adam Gołębiowski, Karolina Dzwonek, Paweł Dobrzański, Zbigniew Zasłona

**Affiliations:** Molecure SA, 02-089 Warsaw, Poland

**Keywords:** arginase, immunotherapy, drug therapy, combination, tumour microenvironment, drug evaluation, preclinical

## Abstract

**Simple Summary:**

Arginase 1 and 2 are drivers of multiple immunosuppressive mechanisms and tumour-specific metabolic adaptations. Pharmacological inhibition of extracellular ARG1 has shown antitumour efficacy in various syngeneic tumour models, however, the importance of ARG2 as a therapeutic target has only been demonstrated by genetic deletion studies. This is the first study validating the benefits of pharmacological inhibition of ARG2 in cancer. Our work describes OATD-02 as a potent dual ARG1/ARG2 inhibitor with a cellular activity (necessary for targeting ARG2) exhibiting immunomodulatory and direct antitumour efficacy in animal models. Our results present OATD-02 as an attractive option for combination with other immunotherapeutics, such as PD-1/PD-L1 antibodies or IDO1 inhibitors, especially in the therapy of particularly resistant hypoxic tumours. The presented findings provided the rationale for planning first-in-human clinical trials for OATD-02 in cancer patients.

**Abstract:**

Background: Arginases play essential roles in metabolic pathways, determining the fitness of both immune and tumour cells. Along with the previously validated role of ARG1 in cancer, the particular significance of ARG2 as a therapeutic target has emerged as its levels correlate with malignant phenotype and poor prognosis. These observations unveil arginases, and specifically ARG2, as well-validated and promising therapeutic targets. OATD-02, a new boronic acid derivative, is the only dual inhibitor, which can address the benefits of pharmacological inhibition of arginase 1 and 2 in cancer. Methods: The inhibitory activity of OATD-02 was determined using recombinant ARG1 and ARG2, as well as in a cellular system using primary hepatocytes and macrophages. In vivo antitumor activity was determined in syngeneic models of colorectal and kidney carcinomas (CT26 and Renca, respectively), as well as in an ARG2-dependent xenograft model of leukaemia (K562). Results: OATD-02 was shown to be a potent dual (ARG1/ARG2) arginase inhibitor with a cellular activity necessary for targeting ARG2. Compared to a reference inhibitor with predominant extracellular activity towards ARG1, we have shown improved and statistically significant antitumor efficacy in the CT26 model and an immunomodulatory effect reflected by Treg inhibition in the Renca model. Importantly, OATD-02 had a superior activity when combined with other immunotherapeutics. Finally, OATD-02 effectively inhibited the proliferation of human K562 leukemic cells both in vitro and in vivo. Conclusions: OATD-02 is a potent small-molecule arginase inhibitor with optimal drug-like properties, including PK/PD profile. Excellent activity against intracellular ARG2 significantly distinguishes OATD-02 from other arginase inhibitors. OATD-02 represents a very promising drug candidate for the combined treatment of tumours, and is the only pharmacological tool that can effectively address the benefits of ARG1/ARG2 inhibition. OATD-02 will enter clinical trials in cancer patients in 2022.

## 1. Introduction

Cancer immunotherapy has become an effective therapeutic strategy for many types of tumours [1]. In recent years, numerous reports have provided evidence that the tumour microenvironment (TME) alters immune cells and converts them into suppressive cells [2,3,4,5]. Altered signalling pathways can impair the adaptive immune mechanisms that eradicate tumour cells. Amino acid metabolism plays a substantial role in the regulation of the immune response [6]. Hence, amino acid-degrading enzymes that are substantially increased in cancer are of particular interest, i.e., indoleamine 2,3-dioxygenase 1 (IDO)-degrading l-tryptophan and arginases hydrolysing l-arginine [7]. Moreover, quantification of interstitial fluid metabolites in murine tumours has revealed that l-arginine is the most strongly depleted amino acid in the TME [8].

There are two distinct arginase isoforms: cytosolic ARG1, which is highly expressed in the liver and is responsible for ammonia detoxification in the urea cycle, and mitochondrial ARG2, which is widely expressed and primarily linked to polyamine generation [9]. The main sources of ARG1 in the TME are myeloid cells that, due to their immunoregulatory functions, are referred to as myeloid-derived suppressor cells (MDSCs). MDSCs express immune checkpoint molecules and release immunosuppressive and proangiogenic factors supporting tumour progression [10]. ARG1-expressing MDSCs were also shown to induce regulatory T cells (Tregs) in mice and cancer patients [11]. Other myeloid cells were also shown to produce ARG1, including tumour-associated macrophages (TAMs) and neutrophils, however, there are considerable differences between mice and men [12]. Many studies confirmed that arginases can be produced by tumour cells [13,14], and when packed into extracellular vesicles (EVs), the enzymes might be transported over long distances and potentially internalized in tumour-draining lymph nodes [15].

Recently, ARG2 attracts particular attention. Similarly to ARG1, ARG2 exerts immunosuppressive effects by inducing l-arginine depletion in the extracellular milieu, as demonstrated using dendritic cells exhibiting deregulated ARG2 expression [16]. Moreover, its expression in neoplastic cells, as well as cancer-associated fibroblasts (CAFs) was correlated with malignant phenotype and poor prognosis [17]. ARG2 was also found to play a critical role in obesity-associated pancreatic cancer (PDA) [18]. Because of highly fibrotic nature, pancreatic tumours are poorly vascularized and generate highly hypoxic and nutrient-deprived TME, which requires PDA-specific metabolic adaptations, such as augmented protein catabolism resulting in ammonia accumulation. ARG2 was revealed to be involved in nitrogen detox and potentially upregulated synthesis of polyamines in highly proliferative PDA tumours [18]. ARG2 has also been implicated in the regulation of metabolism and fitness of T cells [19,20]. ARG2 was found in Tregs from normal skin and its expression increased in metastatic melanoma [19]. Inhibition of ARG2 increased mTOR signalling, whereas overexpression of this enzyme suppressed it, suggesting that Tregs express ARG2 in human tissues to regulate inflammation and enhance their metabolic fitness [19]. Furthermore, ARG2 has been demonstrated as an autonomous regulator of CD8^+^ T cells, affecting their antitumor cytotoxicity and memory formation independently of the extracellular l-arginine availability [20]. Importantly, accumulating data indicate that immune checkpoint inhibitors’ (ICIs) long-term clinical effects are linked with the presence in the TME of the memory-like CD8^+^ T cells, which have the ability to self-renew and develop into effector T cells [21].

The relevance of ARG1 and ARG2 in cancer, documented by many research groups, has prompted a search for pharmaceutical inhibitors of these enzymes in order to alter the outcome of the immune response and develop new treatment options [22]. Several natural and synthetic compounds have been evaluated in various tumour models [22,23], of which CB-1158 (numidargistat), predominantly targeting the extracellular ARG1, is the most advanced in clinical development [24]. The design of competitive arginase inhibitors poses a significant challenge due to the deep and narrow shape of its catalytic pocket. Essentially, all of the inhibitors reported to date are based on their structural resemblance to l-arginine: they are α-amino acids that contain a functional group in their side-chain, capable of chelating two manganese ions necessary for the enzymatic activity. Early studies led to the discovery of 2-(*S*)-amino-6-boronohexanoic acid (ABH), a moderately active arginase inhibitor (IC_50_ = 1 mM) [25] with a boronic acid moiety replacing the guanidine group present in l-arginine. The development of arginase inhibitors was initially focused on improving their inhibitory activity. The approach based on the complementation of the ABH molecule with a fragment that would have the ability to provide additional interactions with Asp181, Asp183, and Asp202 residues was adopted by Van Zandt et al. [26] and later improved by others [27,28]. Further optimization of the potency and overall pharmacological and pharmacokinetic profile was brought about by limiting the conformational flexibility of the ABH molecule. Both CB-1158 (numidargistat) and OATD-02 (patent no. WO2017191130A4), as well as compounds developed by Arcus Biosciences (patent no. WO2019173188A1), Merck Sharp & Dohme (Kenilworth, NJ, USA, patent no. WO2019177873A1), and AstraZeneca (Cambridge, UK, patent no. WO2020161675A1) fall into the category of conformationally constrained ABH analogues. Given the significant differences in pharmacological activities between OATD-02 and CB-1158 inhibitors, we used the latter compound as a reference, highlighting the potential clinical advantages of simultaneous ARG1/ARG2 inhibition.

OATD-02, which differs from CB-1158 in arginase selectivity, overall potency, and PK/PD profile, showed superior antitumor activity in our in vivo studies, which we attribute to the complex action of the compound towards both extracellular and intracellular enzymes. Finally, we are convinced that OATD-02 is the only preclinical candidate capable of addressing the beneficial inhibition of cellular ARG2 that regulates the activity of CD8^+^ cells and Tregs, and constitutes one of the key components of metabolic adaptations of hypoxic tumours along with desired ARG1 inhibition within the TME.

## 2. Materials and Methods

### 2.1. Chemical Compounds

OATD-02 was synthetized at Molecure SA (formerly OncoArendi Therapeutics SA, Warsaw, Poland, patent no. WO2017191130A4) and dissolved in Milli-Q water (Millipore, Burlington, MA, USA) for biochemical and cell-based assays or in sterile saline for in vivo studies. The reference arginase inhibitor (ref. ARGi, INCB001158, numidargistat) was purchased from ChemieTek (Indianapolis, IN, USA) and prepared for the in vivo studies in the same way as OATD-02. All other chemical reagents were purchased from Merck (Darmstadt, Germany) unless indicated otherwise. The IDO1 inhibitor (epacadostat, INCB024360) was purchased from Shanghai Sunshine Chemical Technology (lot SSC151112) and used for oral gavage as a formulation containing 2% DMSO and 18% PEG400 in sterile saline.

### 2.2. Recombinant Arginase Inhibition Assays

Recombinant enzymes (ARG1 or ARG2) were produced at Molecure SA—biosynthesized in *E. coli* expression system, purified by fast protein liquid chromatography, and stored at −80 °C in the storage buffer (20 mM Tris pH 8.0, 100 mM NaCl, 10 mM DDT, 10% glycerol). The enzymatic reaction was carried out at 37 °C for 1 h in the assay buffer (100 mM sodium phosphate buffer, 130 mM NaCl, pH 7.4) with the addition of 1 mg/mL BSA in the presence of 200 µM MnCl_2_ (enzyme cofactor), 10 mM (for ARG1) or 20 mM (for ARG2) of l-arginine hydrochloride (enzyme substrate), and serial dilutions of the test compounds. The inhibition of the arginase activity was determined by colorimetric detection of produced urea, as previously described [29] with minor changes. Briefly, 150 µL of freshly prepared developing reagent (2 mM *o*-phthaldialdehyde (OPA), 2 mM *N*-(1-naphthyl)ethylenediamine dihydrochloride (NED), 50 mM boric acid, 1 M sulfuric acid, 0.03% Brij-35) was added to each well, incubated for 20 min at RT, then the absorbance at 515 nm was measured. Normalized values were fitted to a four-parameter equation using GraphPad Prism and IC_50_ values were determined.

### 2.3. Cell-Based Arginase Inhibition Assays

The intracellular activity of OATD-02 arginase inhibitor was determined using murine M2-polarized macrophages. Bone marrow cells were collected from femur bones of C57BL/6 male mice. The cells suspended in DMEM medium (ThermoFisher, Waltham, MA, USA) supplemented with 5% FBS and 50 ng/mL M-CSF (PeproTech, Rocky Hill, NJ, USA) were poured through a 40 µm cell strainer and seeded into Petri dishes at 1 × 10^6^ cells/mL for 2 days (37 °C, 5% CO_2_). Next, nonadherent cells were removed and fresh medium was added (DMEM, 5% FCS, 50 ng/mL M-CSF). The next day, the cells were dissociated using the CellStripper Dissociation Reagent (Corning, New York, NY, USA) and resuspended in the fresh DMEM medium containing 5% FCS, 50 ng/mL M-CSF, 20 ng/mL TGF-ß, and 30 ng/mL IL-4 (Biomibo, Warszawa, Poland). The cells were seeded into 96-well plates (64 × 10^3^ cells/100 µL/well). Afterwards, the medium was replaced with 80 µL of OptiMEM (ThermoFisher) and 10 µL of serial dilutions of the test compounds and 10 µL of the mixture of l-arginine hydrochloride (5 mM final conc.) and MnCl_2_ (5 µM final conc.) were added. Following 24 h incubation (37 °C, 5% CO_2_), the generated urea was detected by mixing 50 µL of the supernatant with 75 µL of the developing reagent (as described in the above section). Normalized absorbance values (515 nm) were fitted to a four-parameter equation using GraphPad Prism and IC_50_ values were determined. For the liver arginase inhibition assay, human cryopreserved hepatocytes (cat. no. HMCPQC, ThermoFisher/Gibco™) were thawed in the recovery medium (#CM7000, ThermoFisher/Gibco™), supplemented with the Primary Hepatocyte Thawing and Plating Supplements (#CM3000, ThermoFisher/Gibco™), and seeded onto collagen-coated 96-well plates at 9 × 10^4^ cells per well. After 6 h incubation at 37 °C, the medium was replaced with a pre-warmed William’s E Medium (#A1217601, ThermoFisher/Gibco™) and supplemented with the Primary Hepatocyte Maintenance Supplements (#CM4000, ThermoFisher/Gibco™). The next day, the medium was replaced with 80 µL of the OptiMem Reduced Serum Medium (#11058021, ThermoFisher/Gibco™). Serial dilutions of the tested compound were added at the volume 10 µL. Then, a mixture of substrate (l-arginine, 5 mM final conc.) and MnCl_2_ (0.5 µM final conc.) was added at the same volume. The cells were incubated for 24 h and then the supernatants were collected. The urea detection was performed by mixing 50 µL of the supernatant with 75 µL of the developing reagent (as described in the above section). Normalized absorbance values (515 nm) were fitted to a four-parameter equation using GraphPad Prism and IC_50_ values were determined.

### 2.4. Cell Culture

CT26.WT (CRL-2638, CT26, mouse colon carcinoma) and K562 (CCL-243, human CML) cell lines were purchased from the ATCC^®^. Adherent CT26 and non-adherent K562 cells were maintained in the RPMI-1640 medium (Gibco, Life Technologies, Carlsbad, CA, USA) supplemented with 10% foetal bovine serum (FBS, #10270106, ThermoFisher/Gibco™) and 100 U/mL penicillin G and 100 µg/mL streptomycin (Antibiotic-Antimycotic, #15240062, Gibco^®^) at 37 °C in a humidified 5% CO_2_ atmosphere. Full-grown monolayers of CT26 cells were trypsinized for 15 min (0.25% Gibco^®^ Trypsin-EDTA), harvested, and passaged several times for expansion. Both cell lines were confirmed at the last used passage to be mycoplasma-free (Mycoalert Detection Kit, Lonza, Basel, Switzerland). The Renca (murine renal adenocarcinoma) cell line was maintained for the in vivo study conducted at Crown Biosciences (Beijing, China). A monolayer culture was grown in DMEM medium supplemented with 10% FBS at 37 °C and 5% CO_2_. The cells were harvested in exponential growth phase and used for inoculation. For the long-term culture of OATD-02-treated K562 cells, the cells were seeded at a density of 1 × 10^6^ cells/mL on a 24-well plate in a fully-supplemented RPMI-1640 medium and cultured at 37 °C and 5% CO_2_ for two weeks. To assess the antiproliferative effects of OATD-02, the culture medium has not been replaced during this period and the cell density/viability (propidium iodide staining) was determined daily. The viable cell count was normalized vs. control cultures and plotted as percent cell survival.

### 2.5. ARG1 and ARG2 Detection

ARG1 and ARG2 expression was detected in cell lysates by SDS-PAGE followed by western blotting. K562 cells were lysed with Pierce RIPA Buffer (Thermo Scientific) with the Protease Inhibitor Cocktail (Sigma, Tokyo, Japan, #P8340). Extracted proteins (80 µg/well) were run on 10% acrylamide gel and transferred onto a nitrocellulose membrane for antibody staining and detection. The following primary monoclonal antibodies were used: rabbit anti-ARG1 (1:1000, Cell Signaling Technology, Danvers, MA, USA, #93668), rabbit anti-ARG2 (1:1000, Cell Signaling, #55003) and mouse anti-β-tubulin (1:10,000, Millipore, Burlington, MA, USA, 05-661). Also, goat anti-rabbit IgG (1:10,000, Cell Signaling Technology, #7074P2) or horse anti-mouse IgG (1:10,000, Cell Signaling Technology, #7076P2) HRP-conjugated secondary antibodies were used. Detection of the chemiluminescence signal was performed using the Clarity Max Western ECL Substrate (BioRad, Hercules, CA, USA). Before β-tubulin detection, the membranes were stripped with a buffer containing 1.5% glycine, 0.1% DSS and 1% Tween 20 (pH 2.2).

### 2.6. Murine Tumour Models

CT26 and K562 in vivo experiments were conducted at Molecure SA. All experimental procedures were approved by the 1st Local Ethics Committee in Warsaw (Poland). For the syngeneic CT26 model, 7–8-week old BALB/c females (*n* = 9–18 per group, BALB/cAnNCrl, Charles River Laboratories) were injected subcutaneously in the right flank with 5 × 10^5^ CT26 cells. OATD-02, ref. ARGi or epacadostat were dosed by oral gavage twice per day at indicated doses starting 1 day after the tumour implantation. Anti-PD-L1 antibody (InVivoPlus anti-mouse PD-L1, clone 10F.9G2, BioXCell, lot 615416D1) was injected intraperitoneally at 2.5 mg/kg on days: 8, 10, 12, 14 and 16 post tumour cells inoculation. Control groups received vehicle (saline) or isotype antibody (InVivoPlus rat IgG2b, anti-KLH, BioXCell, lot 629816D1). For the xenograft K562 model, 7–8 week-old athymic nude mice (*n* = 8–11 per group, Crl:NU(NCr)-*Foxn1^nu^*, Charles River Laboratories) or NK cell-deficient Fox Chase SCID Beige mice (*n* = 10–12 per group, CB17.Cg-*Prkdc*^scid^Lyst^bg-J^/Crl, Charles River Laboratories) were injected subcutaneously in the right flank with 1 × 10^7^ K562 cells. OATD-02 or ref. ARGi were dosed by oral gavage twice per day at 50 mg/kg starting from day 7 post-inoculation. Tumour volume measured by a digital calliper (width × length × depth × π/6) was recorded three times per week. Animals were euthanized when tumours were necrotized or volume exceeded 2000 mm^3^. Renca in vivo efficacy study was performed at Crown Biosciences Inc. (Taicang, China). All experimental procedures on animals were approved by the Institutional Animal Care and Use Committee (IACUC) and conducted in accordance with the regulations of the Association for Assessment and Accreditation of Laboratory Animal Care (AAALAC). For the syngeneic Renca model, 6–7 week-old BALB/c females (*n* = 12 per group, Shanghai Lingchang Biotechnology, Shanghai, China) were injected subcutaneously in the right flank with 1 × 10^6^ Renca cells. Tumour volumes (length × width^2^ × 0.5) were measured twice per week. OATD-02 was dosed twice per day at 75 mg/kg by oral gavage. A dosing holiday was given to individuals with bodyweight loss exceeding 15% of the initial body weight and the treatment was resumed when the bodyweight loss recovered to less than 10%. Due to the cachexic nature of the Renca model, all animals were supplemented with DietGel to help maintain their body weight.

### 2.7. Flow Cytometry

Dissected tumours were dissociated to obtain single-cell suspensions for antibody staining and immune cell assessment. Prior to the antibody staining, dead cells were labelled with the Fixable Viability Due eFluor™ 780 (eBiosciences, San Diego, CA, USA). The following anti-mouse antibodies were used for flow cytometry: anti-CD45-FITC (clone 30-F11, Biolegend, San Diego, CA, USA), anti-CD3-BUV395 (clone 145-2C11, BD), anti-CD4-BV421 (clone GK1.5, Biolegend), anti-CD8-PE-eFluor610 (clone 53-6.7, eBiosciences), anti-Foxp3-PE (clone FJK-16S, eBiosciences), anti-CD11b-PE-Cy7 (clone M1/70, Biolegend), anti-F4/80-BV510 (clone BM8, Biolegend), and anti-Gr1-APC (clone RB6-8C5, Biolegend).

### 2.8. Bioanalytics

The concentrations of OATD-02, the reference ARGi or l-arginine were determined in plasma samples or tumour homogenates using liquid chromatography coupled with tandem mass spectrometry (LC-MS/MS). Sample preparation was based on protein precipitation using acetonitrile. The resulting supernatants were analysed using hydrophilic interaction liquid chromatography (HILIC) with subsequent MS detection based on optimised MRM transitions. Matrix-matched calibration and quality control samples were prepared for quantification of OATD-02 and the reference ARGi. Due to the endogenous character of l-arginine, a surrogate matrix was used for its detection.

### 2.9. Statistical Analysis

All statistical analyses were performed using GraphPad Prism (ver. 9, GraphPad Software, Inc., San Diego, CA, USA). Data sets with normal distribution were described as means with SEM (standard error of the mean). Otherwise, the median value was shown and the U Mann–Whitney or the Kruskal–Wallis tests were applied for intergroup comparisons. Differences at *p*-value less than 0.05 were considered statistically significant. TGI (tumour growth inhibition) index was calculated by the formula: (1 − A/B) × 100%, where A is the tumour volume in the treated group and B is the volume in the control group.

## 3. Results

### 3.1. OATD-02 Is a Highly Potent Dual ARG1/ARG2 Small-Molecule Inhibitor

OATD-02 was tested in biochemical and cell-based assays to determine its inhibitory activity towards arginases of different origins (Figure 1A,B). A reference arginase inhibitor (ref. ARGi, numidargistat) currently evaluated in clinical trials was used for comparison. OATD-02 strongly inhibited both arginase isoforms (IC_50_: 17 nM ± 2 nM for ARG1, 34 nM ± 5 nM for ARG2) and—in contrast to the ref. ARGi with a predominant activity towards ARG1 (IC_50_: 69 nM ± 2 nM for ARG1, 335 nM ± 32 nM for ARG2)—OATD-02 proved to be a potent dual ARG1/ARG2 inhibitor. Bone marrow-derived M2-polarized macrophages were used to surrogate the myeloid suppressor cells infiltrating the tumour microenvironment. The difference in activities of the tested compounds was even more profound when we evaluated their intracellular activities towards arginase-expressing immune cells (IC_50_: 1.1 µM ± 0.4 µM for OATD-02 vs. 56.2 µM ± 19.0 µM for ref. ARGi) (Figure 1C). Importantly, the inhibition of the intracellular arginase was low for both compounds tested on human primary hepatocytes (IC_50_: 14.2 µM ± 3.4 µM for OATD-02 vs. 50.1 µM ± 7.0 µM for ref. ARGi). Given that the excessive inhibition of the urea cycle in the liver might lead to ammonia-related toxicity issues, these results indicate a similar safety profile of OATD-02 to the ref. ARGi (Figure 1D) being at the same time much more effective in the inhibition of intracellular arginases.

### 3.2. OATD-02 Showed Superior Activity in Combination with ICI and IDO Inhibitor

OATD-02 was tested in the syngeneic CT26 model of colorectal carcinoma along with the ref. ARGi (Figure 2). Both compounds were dosed twice daily at 100 mg/kg by oral gavage, starting from day 1 post-inoculation. We have shown that monotherapy with OATD-02, in contrast to the ref. ARGi with a predominant extracellular activity, significantly inhibited the growth of CT26 tumours, which are characterized by relatively high immune cell infiltration. A statistically significant effect of OATD-02 was observed from day 12 post-inoculation and was sustained until the end of the experiment (Figure 2A, final TGI 48%, *p* = 0.0033 for OATD-02 vs. final TGI 28%, *p* = 0.4390 for ref. ARGi). Despite higher oral bioavailability of the ref. ARGi (Figure 2B), OATD-02 exhibited higher pharmacodynamic effects by increasing the concentrations of endogenous plasma l-arginine up to 6.7-fold vs. 3.3-fold change for the ref. ARGi, as measured at 6 h after the last dosing. These observations were reflected in the tumours (Figure 2C)—within the first 6 h post-dosing, OATD-02 caused a significant increase of l-arginine (up to 6.9-fold vs. 3–6-fold change for ref. ARGi) detected in tumour homogenates. These characteristics represent optimal drug-like properties of OATD-02 at this regiment of dosing. Encouraged by these results, we tested the antitumor activity of OATD-02 in combination with other immunotherapies, specifically epacadostat and anti-PD-L1 antibody. Epacadostat is an investigational, highly potent and selective oral inhibitor of the IDO1 enzyme (indoleamine 2,3-dioxygenase 1). IDO1 contributes to immune suppression by catalysing tryptophan breakdown to kynurenine affecting T cell activation and effector functions [30,31]. In clinical studies, epacadostat combined with immune checkpoint inhibitors has shown limited efficacy in patients with unresectable or metastatic melanoma [32] and non-small cell lung cancer [33], and was further evaluated in patients with advanced solid tumours [34]. Although immune checkpoint inhibitors have revolutionised the treatment of many cancers, resistance to PD-1/PD-L1 blockade remains a significant challenge, necessitating further research into other immune targets [35]. We hypothesized that combining arginase inhibition with the blocking of other immunosuppressive mechanisms, including the expression of immune checkpoints and depletion of nutrients essential for T cell effector functions [36,37,38], will result in potentiated immune-dependent antitumor effects. As shown in Figure 3, CT26 tumour-bearing mice were treated with suboptimal doses of OATD-02, epacadostat or anti-PD-L1 antibody, which led to a partial inhibition of the tumour growth (TGI: 33% for OATD-02, *p* = 0.0042 vs. controls; 41% for epacadostat, *p* = 0.035 vs. controls; 33% for anti-PD-L1 antibody, *p* = 0.0462 vs. controls). Although the dual therapy with epacadostat and anti-PD-L1 antibody did not significantly improve the monotherapies, the addition of OATD-02 led to a superior effect with 87% TGI (*p* = 0.0042 vs. group treated with both epacadostat and anti-PD-L1) observed for this triple combination.

### 3.3. OATD-02 Altered the TME of ARG2-Positive Renca Tumours

In the next step, we evaluated the activity of OATD-02 in the immunogenic Renca model, which is characterized by a strong infiltration of immunosuppressive Tregs and myeloid populations [39,40,41], as well as ARG2 expression by tumour cells [42]. In this cachexic model, a monotherapy with OATD-02 (75 mg/kg, PO, BID) resulted in partial inhibition of the tumour growth (TGI 31%, Figure 4A), however, this effect has not reached statistical significance. We have observed a strong PD effect (6.6-fold increase in plasma l-arginine 16 h upon last dose, Figure 4B), which correlated with a reversal of the immunosuppression in the tumour microenvironment, as evidenced by decreased levels of Tregs (*p* = 0.0196), MDSCs (*p* = 0.0173) and neutrophils (*p* = 0.0284), as well as by a beneficial increase in the CD8^+^/Treg cell ratio (*p* = 0.0281, Figure 4C). These findings further strengthen recent findings on the immunosuppressive role of ARG2 in the promotion of Treg development.

### 3.4. OATD-02 Showed Direct Antitumour Effect on Human ARG2-Positive K562 Cells

Finally, to determine whether OATD-02 can act independently of the immune response by inhibiting the intracellular ARG2 expressed by human tumour cells, we tested the effects of OATD-02 on the survival of leukemic K562 cells. K562 cells are characterized by a high ARG2 expression, which was confirmed by immunoblotting (Figure 5A and Appendix A). We decided to treat these cancer cells at different concentrations of OATD-02 and assessed the percentage of viable cells for a period of 14 days. As shown in Figure 5B, OATD-02-treated cells were dying faster in a dose-dependent manner. The observed effects are thought to be linked to the inhibition of cell-intrinsic ARG2 activity, which might participate in polyamine synthesis [43] and ammonia detoxification [18]. This direct functional effect of ARG2 inhibition in K562 cells encouraged us to test the antitumor activity of OATD-02 in a xenograft in vivo model (Figure 5C). Athymic mice with severe T cell deficiency were inoculated with K562 cells and we started dosing OATD-02 when tumours reached a size of about 100 mm^3^ (day 6 post-inoculation). We used the ref. ARGi for comparison, as the compound was shown to inhibit ARG2 very weakly, and to possess negligible intracellular activity overall. The experiment was terminated when the tumours exceeded 2000 mm^3^. Treatment with OATD-02 inhibited the growth of xenografts (final TGI 49%, *p* = 0.0172 vs. control), whereas the ref. ARGi had no significant antitumor effect (final TGI 14%, *p* = 0.578 vs. control). Given that the therapeutic effects could be partially dependent on the NK cells, which, similarly to T cells, are sensitive to l-arginine deprivation, the experiment was repeated using T and NK cell-deficient animals (Figure 5D). The outcome of this study confirmed previous findings. The growth of K562 tumours was significantly inhibited (final TGI 47%, *p* = 0.0201 vs. control), implying that OATD-02 arginase inhibitor can directly act on the tumour cells by inhibiting ARG2.

## 4. Discussion

Arginase biology has been studied in multiple contexts, resulting in the identification of ARG1 and ARG2 as therapeutic targets in cancer [18,44,45,46], as well as potential biomarkers for cancer progression [47,48,49,50]. Extensive research has provided strong evidence that dysregulated arginase activity accompanies various types of neoplastic diseases [51], and the inhibition of arginases contributes to the restoration of the effective immune response against cancer [4,5,15,24,42,52]. Moreover, targeting arginases with small-molecule inhibitors was reported to affect the malignant phenotype of cancerous cells through multiple mechanisms, including induction of apoptosis and cell cycle arrest, reduction of invasiveness, or cell migration [5,53,54,55,56,57,58,59]. On the other hand, defective l-arginine synthesis, due to the silencing of argininosuccinate synthase 1 (ASS1), is a common metabolic weakness in cancer and results in an intrinsic dependence on extracellular l-arginine due to an inability to synthesise l-arginine for growth [60,61].

Several synthetic and natural arginase inhibitors have been studied and developed to date [27,51,52,53,54]. The most advanced is CB-1158 (numidargistat) developed by Calithera Biosciences and Incyte. CB-1158 has been shown to inhibit tumour growth in multiple syngeneic models. Since it is known to predominantly inhibit the extracellular ARG1 [24], the pharmacological profile of CB-1158 significantly differs from the clinical candidate OATD-02. Therefore, we decided to use CB-1158 (named as the ref. ARGi) in our studies for comparison purposes, since—similarly to other known arginase inhibitors [62]—this molecule lacks the intracellular activity essential to effectively target not only the extracellular myeloid cell-derived ARG1 [24], but also the intracellular ARG2 that regulates the fitness of immune cells [19,20] and significantly determines the tumour growth [17,18,54,63,64,65].

Over the recent years, we and other groups have shown that the benefits of arginase inhibition include enhanced T cell proliferation in response to in vitro stimulation and a decrease in the activity of immunosuppressive myeloid cells [4,15,24]. All these effects do not allow tumours to escape from the T cell-mediated immune surveillance and cytotoxicity. Increased l-arginine in the TME augments T cell proliferation and cytokine secretion, and affects the formation of immune synapse between T cells and antigen-presenting cells [66]. The main mechanism by which the lack of l-arginine inhibits T cell functions is through downregulation of the CD3ζ chain, which is a critical signalling component of the TCR complex [42,52,67,68]. Previously we have shown that this mechanism can be specifically reversed by our dual ARG1/ARG2 inhibitor [4,15]. NK cells, although less sensitive to low l-arginine levels, are also affected by l-arginine starvation, which affects their proliferation, viability, and cytotoxic activity [69]. Therefore, both innate and adoptive immune mechanisms are strongly affected by ARG1 activity in cancer. Recently, the mitochondrial ARG2 has been implicated in metabolic processes, which allow both cancer and suppressive immune cells to function [18,19,70,71]. These findings broaden our understanding of arginase biology and present dual ARG1/ARG2 inhibitor OATD-02 as the first and only pharmacological compound, which can address the benefits of the inhibition of both arginase isoforms.

While the idea of ARG1 inhibition resulting in raising l-arginine in the TME and mounting effective immune response was demonstrated for CB-1158 [24], OATD-02 broadens these benefits by counteracting the ARG2-driven mechanisms. In the syngeneic Renca model, OATD-02 decreased the infiltration of Tregs leading to beneficial CD8^+^/Treg ratios (Figure 4C)—a crucial factor associated with the response to the immune checkpoint inhibitors [72]. ARG2 expression in Tregs was found to play a role in attenuating mTOR activity, which regulates many aspects of Treg biology [19]. Additionally, environmental hypoxia is known to affect ARG2 expression, and thus, the differential oxygen tensions between a hypoxic tumour and a highly vascularized psoriatic lesion might result in altered ARG2 induction between disease states [19]. Along with Tregs, ARG2 has been implicated in influencing immune functions in a cell-intrinsic fashion in erythroid cells [73] and cytotoxic CD8^+^ T cells [20]. Tumour cells can proliferate continuously, so to eradicate tumours, cytotoxic T cells must be endowed with the ability to persist and divide. The longevity of cytotoxic T cells has been associated with a better response to ICIs. ARG2 was found to suppress the development of the long-lived CD8^+^ T memory cells while favouring the generation of short-lived lymphocytes [21]. Furthermore, ARG2 deletion in CD8^+^ T cells has a considerable synergistic effect with PD-1 blockade in controlling tumour growth and animal survival, strongly suggesting that ARG2 is a valuable target for T cell-based cancer immunotherapies.

## 5. Conclusions

OATD-02 is a potent and orally-bioavailable small-molecule inhibitor entering phase I of clinical trials for multiple cancers. Although systemic arginase inhibition can be a concern since it has the potential to disturb the urea cycle and affect the ammonia metabolism in the liver, OATD-02 proved to be safe and well tolerated in animal models. OATD-02 is a first-in-class dual inhibitor of arginases, and we, for the first time, reveal the benefits of inhibiting both ARG1 and ARG2 over ARG1 alone (Figure 6). As of today, we have shown that OATD-02 monotherapy had antitumor efficacy in multiple syngeneic tumour models, including CT26 colorectal adenocarcinoma, Lewis lung carcinoma, B16F10 melanoma, Renca renal adenocarcinoma, ID8 ovarian carcinoma, and GL261 glioma [4,5,15,22], confirming the immunomodulatory action in vivo. Additionally, we have shown that OATD-02 potentiates the antitumor effects of anti-PD-1 [5] and anti-PD-L1 antibodies, epacadostat (IDO inhibitor), and DMXAA (STING agonist) [4]. OATD-02 also effectively inhibited the growth of human cancer cells in the xenograft K562 model, indicating that the therapeutic effects of OATD-02 may be independent of immune system modulation and linked with disrupting hypoxic tumour metabolism (Figure 6). Based on the mapping of ARG2 expression in human cancer (https://www.proteinatlas.org/ENSG00000081181-ARG2/pathology, accessed on 9 August 2022) and literature data, we might anticipate that patients with prostate [65,74], renal [42], and pancreatic cancers [17,18], as well as chronic and acute myeloid leukaemia [13,54], could particularly benefit from ARG2 inhibition. Moreover, the results indicate that OATD-02 can also be effective in the treatment of immunologically cold tumours. All of these independent effects observed in various animal models provide solid foundations for anticipating benefits in cancer patients. The most striking results were obtained for OATD-02 in triple combination therapy with anti-PD-L1, giving a compelling rationale for combinatorial application of arginase inhibitors in the clinical development.

## Figures and Tables

**Figure 1 cancers-14-03967-f001:**
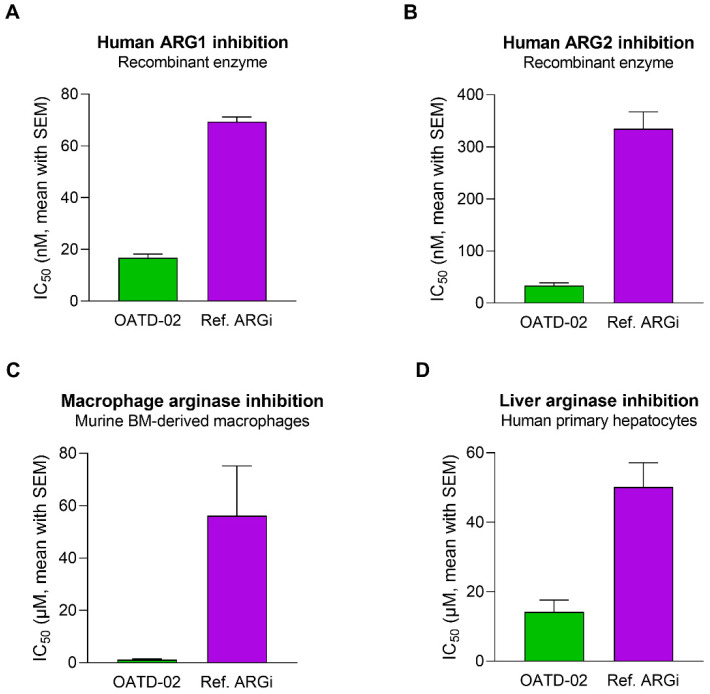
OATD-02 is a highly potent ARG1 and ARG2 inhibitor. In vitro enzymatic assays using recombinant human ARG1 (**A**) and ARG2 (**B**) enzymes mimic the extracellular activity of OATD-02, while cell-based assays performed with murine M2-polarized BMDMs (**C**) and platable human primary hepatocytes (**D**) reveal its potential in inhibiting intracellular arginases. Reference ARG inhibitor (Ref. ARG1) was used for comparative purposes.

**Figure 2 cancers-14-03967-f002:**
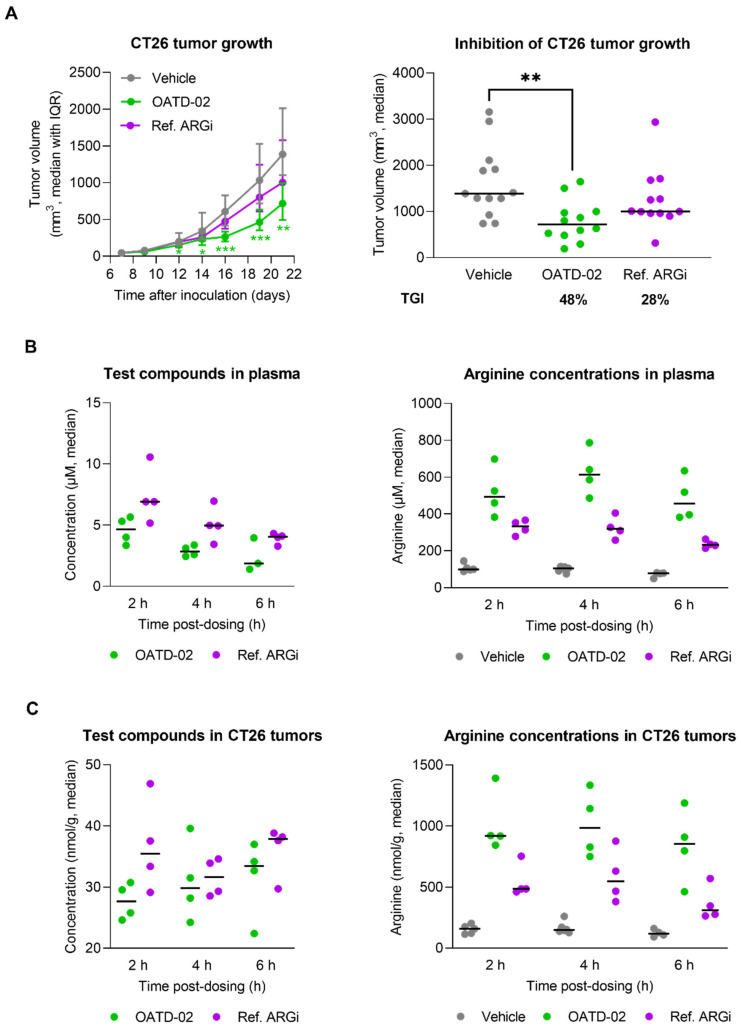
OATD-02 significantly reduces the growth of syngeneic CT26 tumours (colorectal carcinoma) in immunocompetent mice. (**A**)—BALB/c mice were inoculated with CT26 cells and upon 24 h dosed with OATD-02 (100 mg/kg, PO, BID) or Ref. ARGi (100 mg/kg, PO, BID) till the end of the experiment; the final tumour measurements were shown in the right graph (* 0.0281 < *p* < 0.0303, ** *p* = 0.0033, *** 0.0007 < *p* < 0.0008; Kruskal–Wallis test with Dunn’s multiple comparisons test). (**B**)—the concentrations of test compounds in plasma and corresponding l-arginine concentrations. (**C**)—the concentrations of test compounds in tumour homogenates and corresponding l-arginine concentrations.

**Figure 3 cancers-14-03967-f003:**
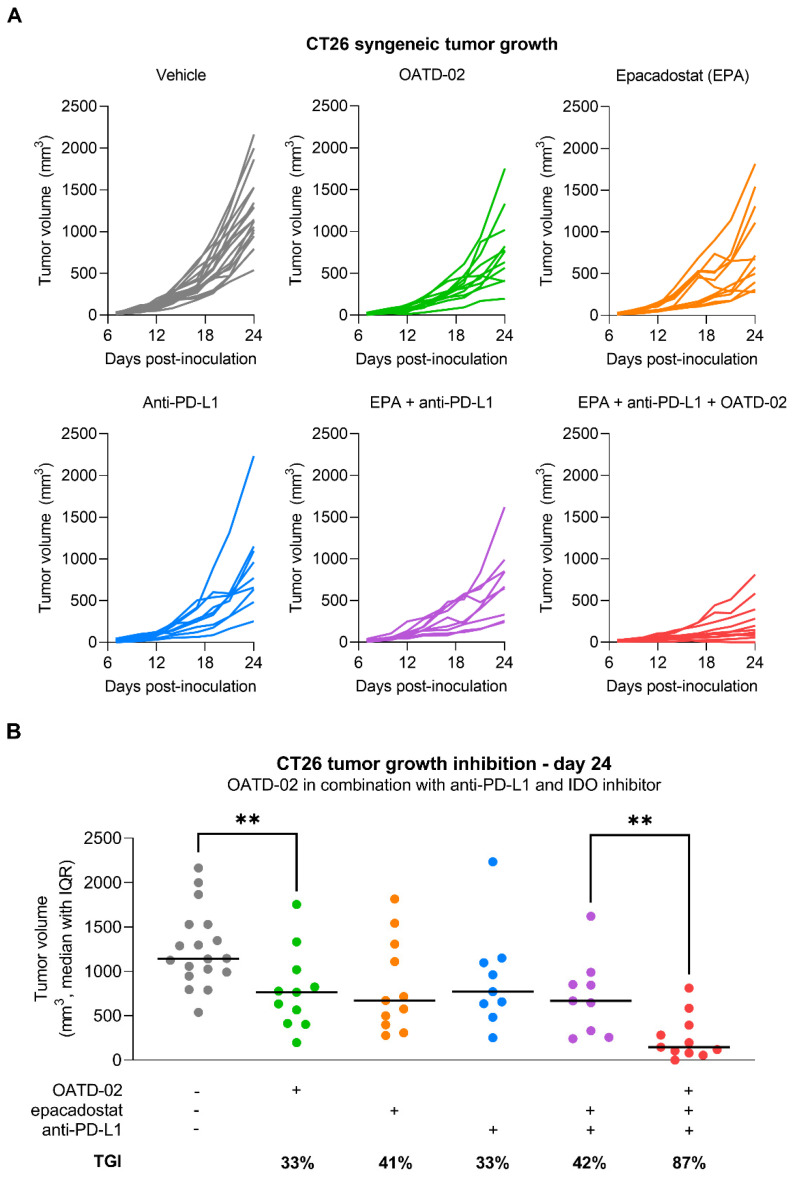
OATD-02 shows superior efficacy in the CT26 tumour model when combined with immune checkpoint inhibitor (anti-PD-L1 antibody) and IDO inhibitor (epacadostat). (**A**)—BALB/c mice were inoculated with CT26 cells and upon 24 h dosed with OATD-02 (50 mg/kg, PO, BID) or epacadostat (30 mg/kg, PO, BID) till the end of the experiment; anti-PD-L1 antibody was administered at 2.5 mg/kg (IP, QD at days: 8, 10, 12, 14 and 16); (**B**)—final measurements of tumour volumes were taken at day 24 post-inoculation and TGI was calculated (** 0.0042 < *p* < 0.0060, U Mann–Whitney test).

**Figure 4 cancers-14-03967-f004:**
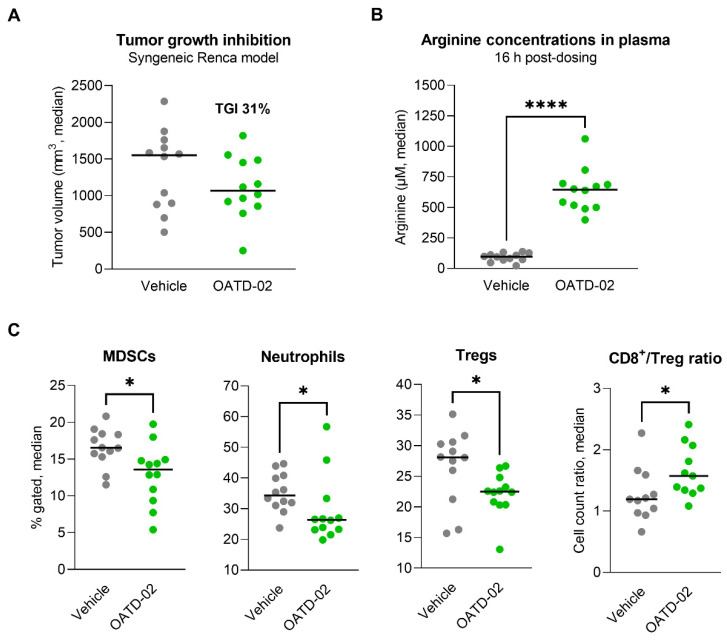
OATD-02 exhibits moderate efficacy in the cachexic Renca tumour model. (**A**)—Renca tumour cells were inoculated at day 0 and OATD-02 was dosed at 75 mg/kg (PO, BID) from day 1 through the end of the experiment; final tumour measurements were taken at day 22 post-inoculation (TGI 31%, not statistically significant). (**B**)—OATD-02 induced high increase in the l-arginine concentrations in plasma measured 16 h after the last dosing (**** *p* = 0.0001, U Mann–Whitney test). (**C**)—Significant changes in the immune infiltration of the Renca tumours were observed upon OATD-02-treatment (* 0.0173 < *p* < 0.0284, U Mann–Whitney test); cell populations were identified by flow cytometry (MDSCs: Gr1^+/−^F4/80^+/−^ within CD11b^+^ cells, neutrophils: Gr1^++^F4/80^−^ within CD11b^+^ cells, Tregs: Foxp3^+^ within CD4^+^ T cells, CD8^+^ T cells: CD3^+^CD4^−^CD8^+^ cells).

**Figure 5 cancers-14-03967-f005:**
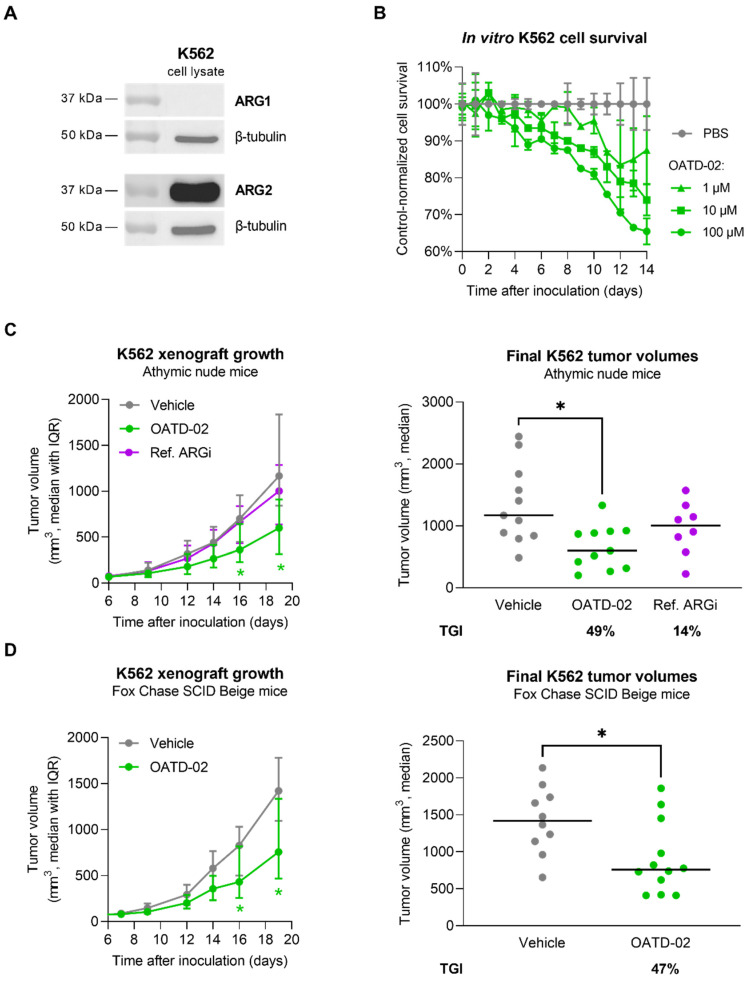
OATD-02 shows significant therapeutic efficacy in the human CML model using K562 cells. (**A**)—ARG1 and ARG2 expression was assessed in K562 cell lysates by SDS-PAGE/WB with specific antibodies; (**B**)—OATD-02 exhibited dose-dependent effect on the cell survival in a long-term K562 cell culture; (**C**)—Athymic nude mice were inoculated with K562 cells and test compound administration started when the xenografts became palpable (day 6); OATD-02 was dosed at 50 mg/kg (PO, BID) and Ref. ARGi was dosed at 100 mg/kg (PO, BID) until the end of the experiment; final tumour measurements were taken at day 19 post-inoculation and TGI was calculated (* 0.0172 < *p* < 0.0428, Kruskal–Wallis test with Dunn’s multiple comparisons test); (**D**)—T and NK cell-deficient mice (strain Fox Chase SCID Beige) were inoculated with K562 cells and OATD-02 was dosed at 50 mg/kg (PO, BID) from day 7 until the end of the experiment; final tumour volumes (day 19) were used for TGI calculations (* 0.0201 < *p* < 0.0449, U Mann–Whitney test).

**Figure 6 cancers-14-03967-f006:**
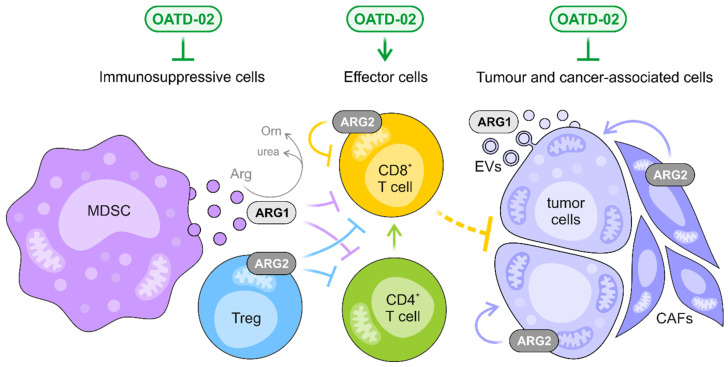
OATD-02 inhibits both extracellular and intracellular ARG1 and ARG2, interfering with multiple immunosuppressive mechanisms, and thus, restoring the effective antitumor immune response. OATD-02 may block the extracellular ARG1 secreted by myeloid-derived suppressor cells (MDSCs) and tumour cells, increasing the availability of l-arginine for effector T and NK cells, but most significantly, the molecule is able to target the intracellular arginases, i.e., cytoplasmic ARG1 and mitochondrial ARG2, that are crucial for the function of immunosuppressive cells, like Tregs, but also negatively regulate the fitness of cytotoxic CD8^+^ T cells. The unique properties of OATD-02 allow it to act on ARG1 packed into the extracellular vesicles (EVs) and counteract ARG2-dependent metabolic adaptations specific for both cancerous and cancer-associated cells, such as cancer-associated fibroblasts (CAFs).

## Data Availability

The data presented in this study are available on request from the corresponding author. The data are not publicly available due to company’s policy.

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
