# Peer review of "OATD-02 Validates the Benefits of Pharmacological Inhibition of Arginase 1 and 2 in Cancer"

_cancers, 2022, doi:10.3390/cancers14163967_

Round 1

Reviewer 1 Report

This paper introduces a study that observed the cancer suppression effect in an animal model using OATD-02, which effectively inhibits ARG1 and ARG2, which are closely related to cancer generation and propagation.  OATD-02 showed superior ARG1 and ARG2 inhibitory effects than the existing arginase inhibitors and effectively inhibited cancer growth.

Comments

As shown in Fig 1., OATD-02 effectively inhibits ARG1 and ARG2. It is recommended to add SDS-PAGE/WB images comparing the levels of ARG1 and ARG2 directly. In addition, Fig. 5a shows the ARG1 and ARG2 expression levels in K562cell. It is recommended to add the SDS-PAGE/WB image that shows decreased ARG expression level after OATD-02 treatment.

 It is difficult to confirm the cancer suppression effect of OATD-02 because of the large standard deviation in Fig. 5. Therefore, it is recommended to clearly explain the inhibitory effect by OATD-02 through the calculation of significance probability.

Author Response

Response to Reviewer 1 Comments

Point 1: As shown in Fig 1., OATD-02 effectively inhibits ARG1 and ARG2. It is recommended to add SDS-PAGE/WB images comparing the levels of ARG1 and ARG2 directly.

 Response 1: We appreciate this comment, however, OATD-02 (similarly to the ref. ARGi) is a reversible arginase inhibitor that does not affect the levels of arginase. Decreased arginase activity is caused by blocking the catalytic site of ARG1 or ARG2, but not by affecting the protein expression. In the assays presented in Fig. 1A and 1B, purified recombinant ARG1 and ARG2 were used and the incubation time was only 60 min.

Point 2: In addition, Fig. 5a shows the ARG1 and ARG2 expression levels in K562cell. It is recommended to add the SDS-PAGE/WB image that shows decreased ARG expression level after OATD-02 treatment.

Response 2: As mentioned above, OATD-02 does not affect the expression of arginase but inhibits its enzymatic activity in a reversible manner. Therefore we expect no difference in protein expression. To be more clear, we have changed the wording in line 378, because the expression of ARG2 by K562 cells is commonly known and described in the literature.

Point 3: It is difficult to confirm the cancer suppression effect of OATD-02 because of the large standard deviation in Fig. 5. Therefore, it is recommended to clearly explain the inhibitory effect by OATD-02 through the calculation of significance probability.

Response 3: Thank the Reviewer’s comment we have modified the text to be more precise and complete. We have added the asterisks to graphs illustrating the kinetics of the tumour growth in Fig. 5A and 5D, as well as updated the figure caption (lines 412-416 and 393-399). To be consistent, we have also corrected Fig. 2A (asterisks) and introduced respective text changes (lines 300-302, 335-336, 348, 369, 371, 393-399 and 412-416). We would also like to emphasize that the cancer suppression effect of OATD-02 was confirmed in two separate xenograft experiments (Fig. 5C and 5D). In both experiments, the final therapeutic effect was assessed when the tumours reached 2000 mm3 in the vehicle-treated groups – the left graphs show the tumour growth kinetics and the graphs on the right show the final tumour measurements. Since a non-parametric statistical analysis was applied, instead of SD, the error bars represent the interquartile range (IQR) which is more informative for data sets with non-normal distribution but indeed does take greater values than SD. The calculation of significance probability is described in section 3.4 (lines 384-392).

Reviewer 2 Report

This manuscript investigates the capacity of a dual inhibitor of arginases 1/2, the OATD-02, to inhibit the tumor growth in several xenografted animal models and compares it with a common arginase 1 inhibitor, ARGi, showing significantly better results that ARGi. As a result, they conclude that OATD-02  is a promising drug candidate for the treatment of cancer. The methods are appropiate and the results are well described and support the conclusion, but I have some minor changes that should be addressed:

- The authors suggest that inhibition of ARG2 (with or without ARG1) is a promising target for cancer treatment and so the demonstrate in the K562 mouse model. However, I would suggest to measure the expression of ARG2 in different cancer tissues discuss which cancer patients could benefit for the dual inhibition of OATD-02.

- Number of animal used should be described in Material and Methods section.

- Is OATD-02 toxic? Does it affect body weight?

Author Response

Response to Reviewer 2 Comments

Point 1: The authors suggest that inhibition of ARG2 (with or without ARG1) is a promising target for cancer treatment and so the demonstrate in the K562 mouse model. However, I would suggest to measure the expression of ARG2 in different cancer tissues discuss which cancer patients could benefit for the dual inhibition of OATD-02.

 Response 1: We fully agree with the Reviewer that the information concerning ARG2 expression is significant. Arginase expression has been extensively mapped by various researchers and is now available for both ARG1 and ARG2 in human cancer in commercial and public databases (https://www.proteinatlas.org/ENSG00000118520-ARG1/pathology, https://www.proteinatlas.org/ENSG00000081181-ARG2/pathology). Therefore, we have referred to a list of cancers which might depend on ARG2, specifically: prostate, renal, pancreatic cancers or chronic and acute myeloid leukaemia. We have expanded this information in the current version of the manuscript (lines 497-501).

Point 2: Number of animal used should be described in Material and Methods section.

Response 2: To address the Reviewer’s comment, we have modified the Materials and methods section to include the number of animals used in vivo experiments (lines 217-236). We would like to point out that the results from the animal studies were presented using dot plot graphs (representing each individual) to provide transparent information about the number of animals used in each experiment and the data spread.

Point 3: Is OATD-02 toxic? Does it affect body weight?

Response 3: In all performed animal studies, the OATD-02 doses were experimentally adjusted to ensure safe drug exposure throughout the dosing period. Each in vivo study, presented in the manuscript, was planned based on the preliminary PK/PD data from the respective mouse strains (BALB/c, athymic nude, Fox SCID Beige). The body weight loss observed in the Renca tumour model occurred in both vehicle- and OATD-02-treated animals. This was due to the cachexia nature of the model and not OATD-02, which is explained in the Material and methods section (line 239-242). Since OATD-02 enters the FIH clinical trials, appropriate safety animal studies were performed in accordance with the recommendations of the European Medicines Agency.

Round 2

Reviewer 1 Report

The queries have been sufficiently addressed. 

 Thanks for addressing all the comments.